# Production in Bacteria and Characterization of Engineered Humanized Fab Fragment against the Nodal Protein

**DOI:** 10.3390/ph16081130

**Published:** 2023-08-10

**Authors:** Jwala P. Sivaccumar, Emanuela Iaccarino, Angela Oliver, Maria Cantile, Pierpaolo Olimpieri, Antonio Leonardi, Menotti Ruvo, Annamaria Sandomenico

**Affiliations:** 1Institute of Biostructures and Bioimaging, CNR, Via P. Castellino, 111, 80131 Naples, Italyemanuela.iaccarino@gmail.com (E.I.);; 2Università degli Studi della Campania Luigi Vanvitelli, Via Vivaldi 43, 81100 Caserta, Italy; 3BIOVIIIX, via A. Manzoni, 1, 80123 Naples, Italy; maria.cantile@bioviiix.com; 4Department of Physics, Sapienza University, 00184 Rome, Italy; pierpaolo.olimpieri@gmail.com; 5Department of Molecular Medicine and Medical Biotechnologies, University of Naples “Federico II”, via Pansini 5, 80131 Naples, Italy

**Keywords:** Nodal, recombinant Fab, humanization

## Abstract

Drug development in recent years is increasingly focused on developing personalized treatments based on blocking molecules selective for therapeutic targets specifically present in individual patients. In this perspective, the specificity of therapeutic targets and blocking agents plays a crucial role. Monoclonal antibodies (mAbs) and their surrogates are increasingly used in this context thanks to their ability to bind therapeutic targets and to inhibit their activity or to transport bioactive molecules into the compartments in which the targets are expressed. Small antibody-like molecules, such as Fabs, are often used in certain clinical settings where small size and better tissue penetration are required. In the wake of this research trend, we developed a murine mAb (3D1) neutralizing the activity of Nodal, an oncofetal protein that is attracting an ever-increasing interest as a selective therapeutic target for several cancer types. Here, we report the preparation of a recombinant Fab of 3D1 that has been humanized through a computational approach starting from the sequence of the murine antibody. The Fab has been expressed in bacterial cells (1 mg/L bacterial culture), biochemically characterized in terms of stability and binding properties by circular dichroism and bio-layer interferometry techniques and tested in vitro on Nodal-positive cancer cells.

## 1. Introduction

Nodal is a potent morphogen belonging to the TGF-β protein superfamily. Its physiological expression is restricted to embryonic stem cells and embryonic tissues where it plays an important role in maintaining embryonic cells pluripotency and in controlling and shaping embryonic and neural development [1,2]. Nodal is rarely expressed in adulthood. Quite an aberrant expression has been instead evidenced in many tumors, including melanoma, prostate, colon, breast, and ovarian carcinomas [3]. Several studies have demonstrated that the over-activation of Nodal signaling is interlinked with the melanoma invasiveness, plasticity, aggressiveness, and tumorigenicity [4,5,6,7,8,9], suggesting that its neutralization may result in tumor suppression.

Nodal exists as both a free ligand in the extracellular/intracellular space and as a component of multimeric complexes on the cellular surface. It works mainly by interacting in a paracrine or autocrine manner with the hetero-multimeric complexes comprising membrane-bound Cripto-1, the activin-like type I (ALK 4/7) and activin-like serine-threonine kinase type II (ActRIIA and ActRIIB, BMPRII) cell surface receptors [3]. It also binds soluble Cripto-1 [10] and some antagonists of the TGF-β receptor family, such as Cerberus, and is able to form covalent heterodimers with other structurally similar ligands, regulating the delicate balance between receptors’ activation/inactivation [11,12]. The interaction with Cripto-1 occurs predominantly with the EGF-like domain, while interaction sites on other proteins are yet to be determined [10]. Binding to these complexes induces the phosphorylation of Smad2/3 that, in turn, associates with the Smad-4. The Smad2/3/4 transcriptional complex translocate to the nucleus where it induces the expression of Mixer and FoxH1, other transcription factors that promote EMT (epithelial-to-mesenchymal transition)-based developmental and oncogenesis progression. Nodal can also interact with ALK4 and ALK7 regardless of the Cripto-1 presence [13,14] and similarly to other ligands belonging to the TGF-β superfamily, has the potential to activate the MAPK/ERK (ras/raf/mitogen-activated protein kinase) and PI3K–AKT signaling. Interestingly, the Nodal-notch intracellular axis has also been proposed as a potential therapeutic target in metastatic melanoma where the re-activation of Nodal expression is under the control of the Notch-4 signal [8].

More recently, it has been involved in the balance between tumor stemness and differentiation [2]. Additionally, Nodal expression has been strictly correlated with the expression of stem cell markers, such as CD44 and CD133, and embryonic transcription factors, such as Sox2, Oct4, and Nanog, in undifferentiated testicular germ cell tumors [15], in breast cancer [16], and in pancreatic CSCs [17].

Finally, Nodal was recently found to induce the expression of L1CAM and CXCR4 in a hypoxic microenvironment. The L1CAM^high^/CXCR4^high^ cell population shows stem-like characteristics and is more tumorigenic and chemo-resistant. Either blocking Nodal or depleting it in these cells may restore the sensitivity towards 5-fluorouracil (5-FU) [18].

Some specific Nodal pathways can be effectively targeted by SB-431542, a potent and specific synthetic inhibitor of ALK4/5/7 receptors [19]. However, given its involvement in multiple pro-survival signaling mechanisms, suppression of its activities requires the inhibition of the very upstream molecular interactions with receptors and co-receptors that trigger signaling propagation.

Based on these considerations and given the active role played by Nodal and Cripto-1 as oncogenes and immunotherapeutic targets for CSC [20], we developed neutralizing mAbs that target specific Nodal [21,22,23] and Cripto-1 [24,25] hot spots mediating the interactions with the receptors involved in their activity. Such mAbs, as well as engineered variants, may become powerful agents for cancer immunotherapy or for specifically accumulating other drugs on cancer tissues and cells expressing them. A previously described anti-Nodal murine monoclonal antibody, named 3D1 against the pre-helix loop of the protein, showed anti-tumor effects in in vitro and in vivo models of melanoma [21,22]. The antibody recognizes specific residues (E49 and E50) of a protein loop involved in the binding to Cripto-1 and prevents their interaction [21,23].

The 3D1 shows therapeutic effects inhibiting the signaling pathways mediated by the two proteins and can potentially work as a carrier for cytotoxic drugs on Nodal-positive cells. One of the crucial steps in generating therapeutic antibody fragments, such as Fabs (fragments antigen binding), is the humanization of the polypeptide chains to suppress immunogenicity. Fabs have a smaller size than whole antibodies and can be more easily humanized. Compared to full antibodies they also show shorter half-life and a superior ability to penetrate the tissue of deep solid tumors and reach hidden epitopes. Additionally, they show faster and extended biodistribution and faster clearance which translate into improved performance for in vivo diagnosis and for short time pharmacological treatments. Finally, the lack of the Fc domain implies a reduced toxicity (complement-dependent cytotoxicity and antibody-dependent cell-mediated cytotoxicity).

Because of these properties, they are the formats of choice for the development of various biotherapeutics, some of which have reached clinical use [26,27].

The possibility of preparing them in bacteria is an added advantage because the process leads to structurally homogeneous non-glycosylated molecules with high yields and reduced preparation time and costs.

Here, we report the preparation of a humanized and ad hoc engineered Fab of the 3D1 antibody that targets the Nodal protein. The humanized Fab bearing at the heavy chain C-terminus a short tag sensitive to transglutaminase-mediated bioconjugations [28], has been recombinantly produced in *E. coli* and characterized for its ability to recognize the Nodal protein in binding assays and for its cytotoxicity in Nodal positive cancer cell lines. Bioconjugation tests have also been successfully performed to confirm the protein ability to be site-specifically modified through a transglutamination reaction, which, by choosing suitable linkers, can be exploited to attach drugs under very mild conditions, avoiding the formation of the heterogeneous products generally obtained through chemical reactions [29].

## 2. Results

### 2.1. Antibody Humanization

Humanization of the murine anti-Nodal mAb 3D1 was obtained through the conventional CDR grafting on the consensus human IgG1 kappa framework of the trastuzumab antibody (see Methods). In Figure 1, the Chothia-numbering scheme for VH and VL variable domains of the murine and CDR-grafted 3D1 is reported. The Fab was built using the humanized variable domain of the parent murine antibody and the constant domain of trastuzumab (https://www.drugbank.ca/drugs/DB00072, accessed on 12 May 2023). The amino acid sequence of humanized Fab of 3D1 (*rh*Fab_3D1) with ad hoc sequence for site-specific modifications is reported in Appendix A.

### 2.2. Expression and Purification of Anti-Nodal rhFab_3D1

The Fab is a heterodimeric and monovalent antibody fragment (50 kDa) composed of an antibody light chain (VL + CL domains) linked by a disulfide bond to the antibody heavy chain (VH + CH1 domains). As the disulfide bridges cannot be efficiently formed in the reducing conditions of the cytoplasm, antibody fragments are most commonly engineered with a signal sequence that directs them to the more oxidizing bacterial periplasm where the presence of enzymes that catalyze the formation, correction, and maintenance of disulfide bonds strongly favors the proper protein folding. The presence of fewer proteases than the cytoplasmatic environment also reduces the risk of degradation during cell growth. As the periplasmatic space accounts for 4–8% of the E. coli protein content, the selective periplasmatic protein extraction on one side reduces the purification efforts, and on the other one reduces the production yield of the recombinant protein. The plasmid pET-26b (+) encoding the engineered *rh*Fab_3D1 was used for expression in the periplasm of the *E. coli* BL21 (DE3) strain. The expression was optimized in SB-MOPS medium with 2 mM IPTG induction overnight at 16 °C at 150 rpm. The pellet was accurately lysed, and the *rh*Fab_3D1 was efficiently and homogenously purified using a CH1-based affinity column with a recovery of about 1 mg per liter of culture. After buffer exchange in PBS, the *rh*Fab_3D1 was analyzed on a Superdex 75/100 column to confirm its monomeric status (Figure 2a). The purified *rh*Fab_3D1 was analyzed on a 15% SDS-PAGE gel (Figure 2b) where the protein sample appeared as a single band at around 50 kDa under nonreducing conditions and as two well-separated bands, corresponding to the light chain (24.3 kDa) and the heavy chain (25.4 KDa), under reducing conditions. LC-ESI-TOF-MS analyses confirmed the identity and integrity of *rh*Fab_3D1 under both nonreducing and reducing conditions (Appendix A and Table 1).

### 2.3. Structural Characterization of rhFab_3D1

Circular dichroism (CD) analyses were performed on the *rh*Fab_3D1 fragment to evaluate its proper folding in terms of secondary structure and stability. The CD spectrum recorded in the far-UV region (Figure 3A) showed that *rh*Fab_3D1 adopts the expected β-sheet-rich secondary structure organization with a negative band at 219 nm and a positive band at 205 nm. The thermal denaturation curve obtained monitoring the CD signal at 218 nm between 20 °C and 95 °C revealed a melting temperature (Tm) of 82 °C (Figure 3B), which is comparable to that measured for other recombinant Fabs [28,30].

### 2.4. Binding of rhFab_3D1 to rhNodal

The affinity of *rh*3D1_Fab to the rhNodal protein was evaluated through bio-layer interferometry (BLI) kinetic binding assays. The recombinant protein Nodal was immobilized onto an AR2G sensor chip, and solutions of the Fab were analyzed at concentrations ranging between 6.25 nM and 100 nM. The kinetic parameters (k_on_ and k_off_) were extrapolated from the binding curves and fitted (Figure 4). The K_D_ was calculated averaging the values obtained from the three highest concentrations (25 nM, 50 nM, 100 nM) since those obtained at low concentrations were too divergent and likely suffered from the high immobilization level of the sensor chip. The estimated K_D_ thus was 7.8 nM (Table 2) which is about a five-fold decrease compared to that of the full-length mAb 3D1 (1.4 nM) [21] and about a two-fold increase as compared to the value reported for the murine 3D1 Fab previously reported, suggesting that the humanization also positively affected the affinity exhibited by the monovalent Fab.

### 2.5. Site-Specific Labeling of Recombinant rhFab_3D1 with Fluorescent Molecules Using Microbial Trasglutaminase (MTG)

Site-specific labeling reactions were performed on the *rh*Fab_3D1 to assess whether under appropriate conditions and with suitable drugs it could be used as a carrier of fluorescent dyes and chemotherapeutic drugs. For this purpose, the peptide linker, FITC-βA-βA-KAYA-CONH_2_ [25,28], bearing a lysine highly sensitive to MTG was used [31]. The reaction was monitored over time by LC-ESI-TOF mass spectrometry and stopped adding 10 mM iodoacetamide to prevent the MTG-mediated reverted reaction of hydrolysis. As shown in Figure 5, after 10 min, more than 95% of product was labeled with FITC-βA-βA-KAYA-NH_2_, as proven by the increase of molecular weight (MW_calculated_ 50,649.28 Da/MW_experimental_ 50,650.95 Da).

### 2.6. In Vitro Cytotoxicity Assay

The ability of the recombinant fragment *rh*Fab_3D1 to inhibit the growth of cells expressing the Nodal antigen was evaluated on the NT2-D1 cancer cells using WST-based assays. *rh*Fab_3D1 was tested at concentrations ranging between 10 nM (0.5 μg/mL) and 5 μM (250 μg/mL). As positive control, a commercial anti-Nodal antibody (WS65) was tested at concentrations ranging between 1 and 100 nM (0.05 μg/mL and 0.5 μg/mL) (Appendix A). As shown in Figure 6, *rh*Fab_3D1 exhibited a dose-dependent growth inhibitory effect, with an average IC_50_ of around 1 μM and a cell death rate of 57% (43% cell viability). The commercial anti-Nodal antibody instead inhibited more potently cell growth, showing an IC_50_ between 50 nM and 100 nM.

## 3. Discussion

Monoclonal antibodies represent an increasingly attractive category of biotherapeutics. In fact, more than 100 mAbs have already been FDA-approved, and many others are being evaluated in clinical trials. However, despite their numerous advantages, such as long in vivo half-life, low aggregation propensity, and high thermal stability, their use in certain diseases is restricted by the large size that limits tissue penetration and access to cryptic or hidden epitopes [32,33]. In order to overcome these drawbacks, next-generation therapeutics based on structurally engineered antibody surrogates, including antibody fragments, with improved pharmacokinetics and pharmacodynamics properties have been realized and some are used clinically [26,27]. Innovative recombinant and multi-engineered antibody-like formats are thus already promising immuno-therapeutics for the treatment of the most widespread diseases, such as cancer, immune disorders, and infectious diseases. Fabs are amongst the most successful antibody fragments approved by the U.S. Food and Drug Administration (FDA) for clinical use (e.g., certolizumab pegol, ranibizumab, abciximab, idarucizumab; https://go.drugbank.com/categories/DBCAT001914, accessed on 12 May 2023), and together with other antibody formats are increasingly being studied. Following this line of development that tends to make antibody molecules smaller in size but with similar biological properties, we tested the feasibility of making a humanized Fab derived from the murine anti-Nodal antibody 3D1 already well-characterized for its ability to neutralize the Nodal activity in in vivo tumor models. Using an established *E. coli* platform, an engineered humanized recombinant Fab has been generated and deeply characterized at the analytical and structural level by SEC, SDS-PAGE, LC-MS, and CD. The affinity for the recombinant human protein has been also determined by BLI, confirming the binding ability of the new molecule to interact with the antigen. Its ability to kill cancer cells expressing the target antigen has been evaluated on NT2-D1 cells. Although at a higher concentration compared to a commercially available anti-Nodal full-length antibody, the humanized Fab reduces cell vitality in a concentration-dependent manner, demonstrating a potential for therapeutic development. The relatively high EC50 still suggests that the molecule needs further optimization in terms of affinity and probably also ability to reach the epitope of the target protein. The data presented here are therefore to be considered preparatory to future studies focusing on the further characterization of the antibody molecule and are only a first indication that the structural changes introduced by mutations do not result in the total loss of biological activity. In order to evaluate its prospect application as a carrier for cancer drugs, we have also investigated its ability to undergo enzymatic bioconjugations using a microbial transglutaminase. This was made possible by the introduction of a consensus sequence sensitive to this covalent modification on the heavy chain C-terminus. Data show that the Fab can be site-specifically modified with a FITC-peptide that mimics a linker with an attached drug. Although the data are somehow preliminary, they clearly suggest that humanized Fab molecules derived from antibodies with high therapeutic potential can be successfully prepared in bacterial platforms and engineered in a way that enables their covalent modification under very mild conditions and in a site-specific manner. One of the problems often encountered with antibody fragments is indeed the poor structural stability that can lead to domain rearrangements, the formation of oligomeric or polymeric structures, and the loss of biological activity. In contrast, our humanized anti-Nodal Fab exhibits a thermal denaturation resistance comparable to that of trastuzumab, a monomeric status, and retains an almost unchanged affinity for the antigen.

The interest in Nodal stems from the role it plays in several types of cancer. Indeed, Nodal together with Cripto-1 is involved in maintaining the embryonic state of CSCs and performs its function by interacting with multiple protein partners and triggering various transduction pathways. The use of monoclonal antibodies or Fabs able to block these interactions are therefore of particular relevance in this context, especially considering that they can also just act as vehicles for cytotoxic or fluorescent agents that allow monitoring, even in vivo, of drug accumulation in tumor masses and disease progression.

It is also important to underline that the presence of the MTG-responsive site opens up other scenarios for the generation of new multimeric and multispecific constructs capable of targeting more than one antigen and interrupting multiple interactions in a single complex [28]. In the case of the receptor complexes involved in Nodal and Cripto-1 signaling, the creation of bispecific Fabs that can bind both molecules and thus more effectively block their tumor growth signals is of great therapeutic interest [3] and is an area of scientific investigation deserving great attention.

## 4. Materials and Methods

The chromatographic columns and AKTA FPLC system were provided by Cytiva (Milan, Italy). Certified reagents used to perform BLI analyses and OCTET R8 instrument were supplied by Sartorius (Göttingen, Germany). Electrophoresis reagents were provided by Bio-Rad (Milan, Italy). Materials used for peptide synthesis, such as protected amino acids, coupling agents (HATU, Oxyma), Fmoc-Rink amide resin, as well as solvents and other products, including acetonitrile (CH_3_CN), dimethylformamide (DMF), trifluoroacetic acid (TFA), diisopropylethylamine (DIPEA), and piperidine, were purchased from Merck/Sigma-Aldrich (Milan, Italy). The peptides were purified using a HPLC WATERS 2545 preparative system (Waters, Milan, Italy). The Nodal recombinant protein (code 3218-ND/CF) was purchased from R&D Systems (Minneapolis, MN, USA). The commercial anti-Nodal antibody WS65 (code sc-81953) was purchased from Santa Cruz Biotechnology (Dallas, TX, USA). The enzyme MTGase (EC. 2.3.2.13) from Streptoverticillium mobaraensis, Activa WM, 81–135 U/g (MTG) was purified as previously reported [27,28]. The Nodal Fab synthetic gene, cloned in the pET-26b (+) vector, was purchased from Genescript (Piscataway, New Jersey, USA). The Nodal overexpressing cells, human teratocarcinoma cell line, N-tera2 D1 (NT2-D1), were obtained from ICLC cell bank (Genova, Italy) and confirmed as authentic and contamination-free. Dulbecco’s modified Eagle’s medium (DMEM) with high glucose formula was supplemented with 10% fetal bovine serum (Gibco, Thermo Fisher, Segrate, Italy). LC-MS analyses on recombinant Fabs and synthetic peptides were performed using an Agilent 1290 Infinity LC System coupled to an Agilent 6230 time-of-flight (TOF) MS System (Agilent Technologies, Cernusco Sul Naviglio, Italy), using C18 (peptides) or C4 (proteins) BioBasic (5 μm, 50 × 2.1 mm) columns, applying linear gradients, as reported in the methods. 0.05% TFA in H_2_O and 0.05% TFA in CH_3_CN were used as solvent A and solvent B, respectively. Flow rate for analytical determinations was 0.2 mL/min.

### 4.1. Computational Approach for Generating the Humanized Recombinant Fab

Sequencing of the monoclonal antibody 3D1 was performed as described previously [25]. The murine amino acid sequences of VH and VL were obtained using the program Translate from the ExPASy proteomic server. The consensus human IgG1 kappa framework of the trastuzumab antibody (https://go.drugbank.com/drugs/DB00072, accessed on 1 May 2023) was used for humanization of the murine anti-Nodal mAb 3D1 via the CDR-grafting approach. Trastuzumab was chosen as it is one of the best-known mAbs and among the first to be placed on the market. It has a highly optimized sequence and is indeed very stable to both temperature changes as evidenced by its resistance to thermal denaturation and aggregation over the long term of both the full antibody and the corresponding Fab [30]. Possible affinity losses on this variant were evaluated using the Tabhu (Tools for antibody humanization) web server [34], which predicted and compared its binding modes compared to the mouse precursor, also highlighting differences in the framework region (FR) residues. On the basis of these results (not shown), a set of back mutations was identified and introduced. The selected mutations, according to the Chothia-numbering scheme [35], were residues 83 (F->L) and 3 (Q->V) of the kappa light chain and residues 71 (A->R) and 94 (R->G) of the heavy chain. Residues 71 and 94 of the heavy chain play a critical role in determining the canonical structure of the H2 and H3 loops, respectively. Therefore, their contribution seems crucial in shaping the antibody-binding site. Residues 3 (Q->V) and 83 (F->L) of the light chain were selected after the structural comparison of three-dimensional models predicted by the PIGS web server of the murine and grafted variable regions [36], integrating the results of the random forest algorithm in Tabhu. Despite its position in the FR, residue 83 of the light chain was selected on the first round of back mutations since the predicted score was one of the highest. The variant obtained after this round of mutations was again compared to the binding modes of 3D1, finding that they were very similar. A subsequent structural analysis performed with Tabhu confirmed that no further back mutations were deemed to be required; therefore, no mutations in the light chain sequence were done.

### 4.2. Design and Expression of Recombinant Humanized Fab

The ad hoc designed gene encoding for the humanized Fab of the 3D1 anti-Nodal antibody (*rh*Fab_3D1) was cloned into the pET-26b (+) expression vector for periplasmatic expression and optimized for the *E.coli* host. As previously reported [25,28], the synthetic gene was opportunely modified at the C-terminus of the heavy chain to encode the peptide sequence GSGALQPT**Q**GAMPA containing a glutamine residue (in bold) sensitive to transglutamination by MTG enzyme (Appendix A). *rh*Fab_3D1 expression was optimized in the BL21(DE-3) *E. coli* strain (Invitrogen) in super broth medium with MOPS buffer pH 7.0 (SB-MOPS) (35 g/L tryptone, 20 g/L yeast extract, 5 g/L NaCl, 10 g/L MOPS). During the pre-induction, the cells were grown in SB-MOPS medium supplemented with 0.2% glucose and Kanamycin (0.03 g/L) in a 37 °C orbital shaker set at a speed of 150 rpm under an aerobic condition. Cells were checked until they reached an OD (optical density) of 0.6 at 600 nm. Cultures were transferred into the fresh glucose-free SB-MOPS medium before induction with IPTG used at 2 mM (isopropyl-β-d-thiogalacto-pyranoside). Cell cultures were incubated for 16 h in an orbital shaker set at 150 rpm and 16 °C. The bacterial cultures were finally harvested by centrifugation, and the resulting pellet was washed with PBS buffer and stored at −80 °C.

### 4.3. Purification and Analytical Characterization

The periplasmic extraction was performed in 100 mM Tris-HCl at pH 7.4, 10 mM EDTA at pH 8.0 containing a protease inhibitor cocktail (Complete EDTA free tablet, Roche, Basel, Switzerland). The mixture was kept overnight at 30 °C in an orbital shaker under constant agitation (250 rpm). After centrifugation at 12,000 rpm for 30 min at 4 °C, the lysate was again clarified by filtration on a 0.2 μm sterile membrane. *rh*Fab_3D1 was purified using a CaptureSelect™ IgG-CH1 affinity column (Thermo Fisher Scientific, Segrate, Italy) according to the manufacturer instructions connected to an AKTA purifier system (Cytiva, Milan, Italy). The Fab was eluted using 0.1 M glycine at pH 2.5. The bound fractions were collected and immediately neutralized with 2 M Tris-HCl at pH 9, buffer exchanged with PBS and concentrated using the Millipore Amicon centricon 30 kDa cut-off (Merck, Darmstadt, German). The oligomeric state of *rh*Fab_3D1 was assessed by gel filtration on a Sephadex 75 column (Cytiva, Milan, Italy) equilibrated with PBS pH 7.0 or 25 mM Tris-HCl, 100 mM NaCl, pH 7.5. The purity was estimated by SDS-PAGE analysis, and the concentration was measured through determination of the absorbance at 280 nm using a NanoDrop 2000C spectrophotometer. The following parameters were used for the calculation: *rh*Fab_3D1 MW 49,714.54 Da; ε_280nm_ 75,915 M^−1^ × cm^−1^).

### 4.4. Structural Characterization through CD Analysis

CD analysis was conducted using a Jasco spectropolarimeter model J-710 equipped with a Peltier system to control the temperature and with a 110-QS quartz cuvette with a 1.0 mm light path length (Hellma; Mullheim, Baden, Germany). Each sample was prepared in 10 mM sodium phosphate, pH 7.5 at 0.2 mg/mL (4 μM). Spectra were collected within the wavelength range of 250–190 nm in the far-UV region at a scan rate of 20 nm/min with a data pitch of 0.2 nm, a bandwidth of 1 nm, and a response time of 4 seconds. A thermal denaturation experiment was performed by monitoring changes in ellipticity at 220 nm during exposure to increasing temperatures between 40 and 90 °C, heating at 1 °C/min. The melting point was determined by the method of the first derivative. Three independent spectra for each sample were recorded, averaged, and smoothed using the Spectra Manager software, version 1.53 (Easton, MD, USA). Final spectra were corrected by subtracting the corresponding baseline spectrum obtained under identical conditions. Data were exported and charted using Graph Pad version 5 software.

### 4.5. Biolayer Interferometry (BLI) Label-Free Binding Assays

The BLI technique was used to assess recognition between *rh*Fab_3D1 and human recombinant Nodal (*rh*Nodal). The protein was efficiently immobilized at 10 μg/mL in 10 mM NaAc pH 4.5 on AR2G sensor chips (Sartorius, Göttingen, Germany) according to the manufacturer’s instructions. A reference channel was used as blank. *rh*Fab_3D1 was tested at increasing concentrations between 6.25 and 100 nM using the kinetic buffer (PBS contain 0.02% Tween20, 0.1% BSA, 0.05% sodium azide) as the running buffer. All analyses were carried out at 25 °C. A 5 mM NaOH solution was used to regenerate the chip surface. All mathematical manipulations and fittings were performed using the Octet Analysis Studio 12.2 from Sartorius. Data were fitted assuming a 1:1 Langmuir binding model considering the monovalency of the Fab. The K_D_ was calculated averaging the values obtained from the 3 highest concentrations (25 nM, 50 nM, 100 nM) since those obtained at low concentrations were too divergent and likely suffered from the high immobilization level of the sensor chip. The term “Error” given in Table 2 indicates for each individual experiment, the point-to-point average deviations calculated between the values of the experimental curves and the values of the curves obtained from fittings. It reflects the goodness of the fittings of the experimental sensorgrams with the theoretical association and dissociation curves. In column 2 of the same table, the values refer to the deviations of the KDs calculated from the deviations of the corresponding k_on_ and k_off_. Binding curves were exported and charted using Graph Pad version 5 software.

### 4.6. Solid-Phase Peptide Synthesis and Purification

Peptides linkers were synthesized as previously reported [25,28,31] following standard Fmoc chemistry protocols using a Rink-amide MBHA resin (substitution 0.56 mmol/g) and amino acid derivatives with standard side chain protections. Synthesis procedures and utilized experimental conditions were optimized and described elsewhere [31,37]. After cleavage (TFA/TIS/water (90:5:5, *v*/*v*/*v*), peptides were purified to homogeneity by RP-HPLC using an X-Bridge Prep C18 column (19 × 150 mm ID), applying a linear gradient of 0.1% TFA in CH3CN from 5% to 70% over 15 min (flow rate at 15 mL/min) and using a Waters LC Prep 150 HPLC system. Peptide purity and identity were assessed by LC-ESI-TOF-MS.

### 4.7. Site Specific Bioconjugations of Linkers via MTG

To assess the versatility of enzymatic reactions to introduce chemical moieties on biomolecules, the anti-Nodal recombinant humanized Fab was subjected to a number of bioconjugations as reported previously with a recombinant fragment of the trastuzumab [28]. MTG *rh*Fab_3D1 was therefore site-specifically labelled at the C-terminus of the Fab heavy chain with the fluorophore-containing peptide FITC-βA-βA-KAYA-CONH_2_ (MW_Exp/Theor_: 951.39 /950.64 amu), monitoring the reaction progression by LC-ESI-TOF mass spectrometry.

### 4.8. Peptides and Protein Identification by LC- ESI-TOF-MS Analysis

Mass spectrometry analyses were performed with an Agilent 1290 Infinity LC System coupled to an Agilent 6230 TOF as reported previously [28].

The liquid chromatographic Agilent 1290 LC module was coupled with a PDA detector and a 6230 time-of-flight MS detector, along with a binary solvent pump degasser, column heater, and autosampler. The pump was connected to a gradient binary solvent system: A, 0.01% TFA/H_2_O (*v*/*v*), and B, 0.01% TFA/CH_3_CN (*v*/*v*). Chromatographic analyses of linker peptides were performed using a reverse phase C18 Biobasic column applying a linear gradient from 5% to 70% of B for 10 min. Chromatographic analyses of recombinant Fabs under nonreducing and reducing conditions were performed using a reverse phase C4 Biobasic column applying a linear gradient from 25% to 65% of B for 15 min. The column flow rate was kept at 0.2 mL/min with the heater at a constant temperature of 20 °C. UV spectra were monitored in the range between 200 and 600 nm. The mass analyzer Agilent 6230 TOF-MS was set to operate in positive ion scan mode with mass scanning from 100 to 3200 *m*/*z*. The instrument acquired data using the following parameters: drying gas temperature, 325 °C; drying gas flow, 10 L/min; nebulizer, 20 psi; sheath gas temperature, 400 °C; sheath gas flow, 11 L/min; VCap. 3.500 V; nozzle, 0 V; fragmentor, 200 V; skimmer, 65 V; and octapole RF Vpp was 750 V. Data collection and integration were performed using MassHunter workstation software (version B.05.00). Data were stored in both centroid and profile formats during acquisition. A constant flow of Agilent TOF reference solution through the reference nebulizer allowed the system to continuously correct for any mass drift by using two independent reference lock-mass ions, purine (*m*/*z* 119.03632), and HP-922 (*m*/*z* 922.000725) to ensure mass accuracy and reproducibility. Target compounds were detected and reported from accurate mass scan data using Agilent MassHunter Qualitative software version B.05.00.

### 4.9. In Vitro Assays

#### 4.9.1. Cell Culture

NT2-D1 cells were cultured in DMEM with high glucose formula supplemented with 10% fetal bovine serum, 2 mM glutamine, and 1% penicillin/streptomycin in an incubator at 37 °C with a humidified atmosphere containing 5% CO_2_.

#### 4.9.2. In Vitro Cell Viability Assays

In vitro cytotoxicity of the Fab was determined by the WST-1 tetrazolium salt colorimetric growth assay. NT2-D1 cells were trypsinized, counted, and seeded at a density of 3 × 10^3^ cells/well on a 96-well plate overnight. The next day, cells were treated with anti-Nodal antibodies at concentrations ranging between 1 and 100 nM (commercial anti-Nodal WS65, positive control) and between 10 nM and 10 µM with the *rh*Fab_3D1. After 72 h, cell viability was estimated adding 1/10 volume of WST-1 solution (Roche, Basel, Switzerland) and incubating for 1 h at 37 °C. The absorbance at 450/650 nm was determined using a microplate reader (BioTek Synergy LX, Agilent, Cernusco sul Naviglio, Italy). Results were expressed as percentage of control. Experiments were performed in three replicate wells and at least thrice. Data are reported as the mean of all experiments with associated standard error values. Curves were fitted using nonlinear regression using Graph Pad version 9.2.

## Figures and Tables

**Figure 1 pharmaceuticals-16-01130-f001:**
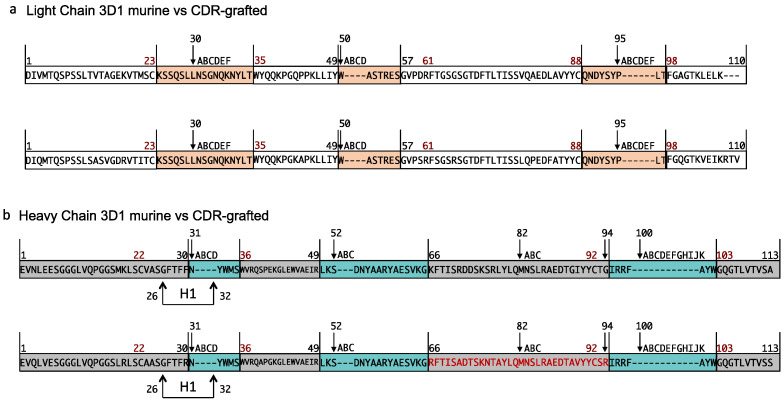
Chothia-numbering scheme for VH and VL of murine and humanized 3D1 Fabs. The numbers above the sequences represent the numbering of specific residues. Complementarity determining regions are colored in light green for VH and in orange for VL. Letters correspond to insertions. Framework regions are depicted in grey for VH and in white for VL. Arrows indicate Chothia and Lesk definition of hypervariable loops.

**Figure 2 pharmaceuticals-16-01130-f002:**
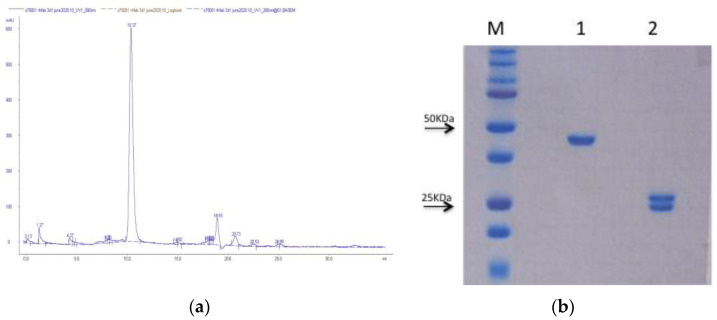
(**a**) Size exclusion chromatogram of *rh*Fab_3D1 obtained with a S75/100 column. The Fab eluted as a single sharp peak at 10.37 mL, corresponding to a molecular weight of around 50 kDa; (**b**) 15% SDS-PAGE gel of the purified *rh*Fab_3D1 under nonreducing (lane 1) and reducing conditions (lane 2). The molecular weight ladder (Dual Color Standards, Bio-Rad) is shown in lane M.

**Figure 3 pharmaceuticals-16-01130-f003:**
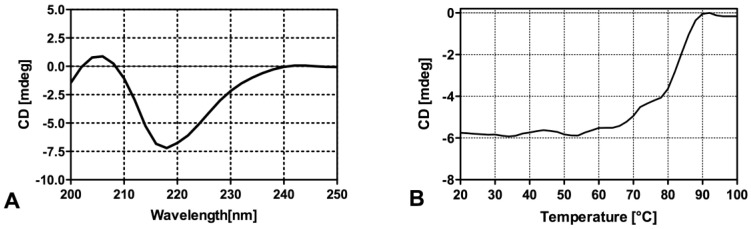
CD spectra of *rh*Fab_3D1 in the far-UV region collected on a protein solution in 10 mM phosphate pH 7.0, 0.4 mg/mL, and 20 °C (**A**). In (**B**) the denaturation curve collected between 20° and 95 °C following the CD signal at 218 nm is reported.

**Figure 4 pharmaceuticals-16-01130-f004:**
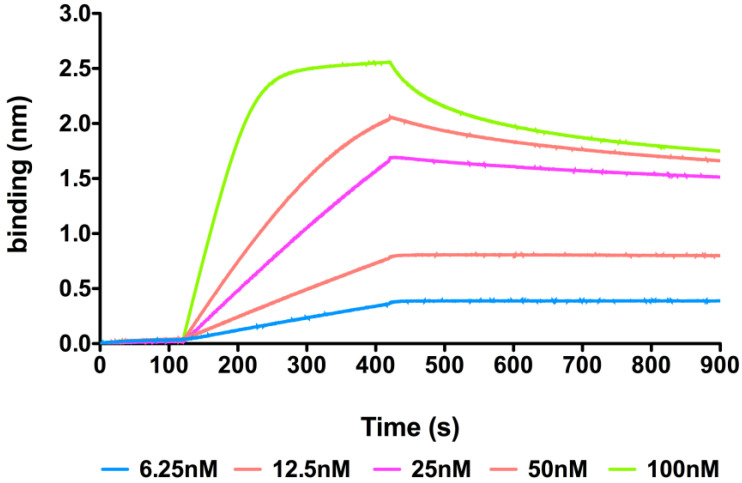
Bio-layer interferometry (BLI) curves obtained for *rh*Fab_3D1 binding to immobilized *rh*Nodal protein on AR2G sensor chips. Only the sensorgrams obtained at 25 nM, 50 nM, and 100 nM were used to extrapolate the K_D_.

**Figure 5 pharmaceuticals-16-01130-f005:**
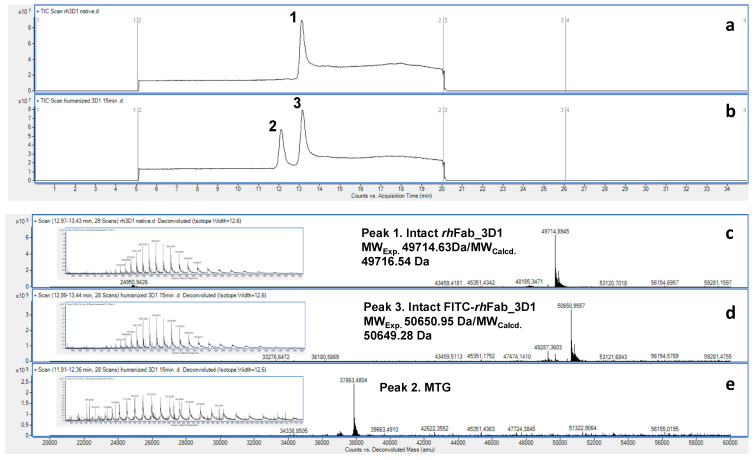
LC-ESI-TOF-MS analyses of *rh*Fab_3D1 before and after MTG-mediated bioconjugation with FITC-βA-βA-KAYA-NH_2_. (**a**) TIC chromatogram of the intact *rh*Fab_3D1 (peak 1) and after 10 min reaction (**b**). Peak 2 corresponds to MTG, while peak 3 accounts for the FITC-modified *rh*Fab_3D1. In (**c**–**e**) the deconvoluted and multicharged (insets) mass spectra of peaks 1, 2, and 3 are reported.

**Figure 6 pharmaceuticals-16-01130-f006:**
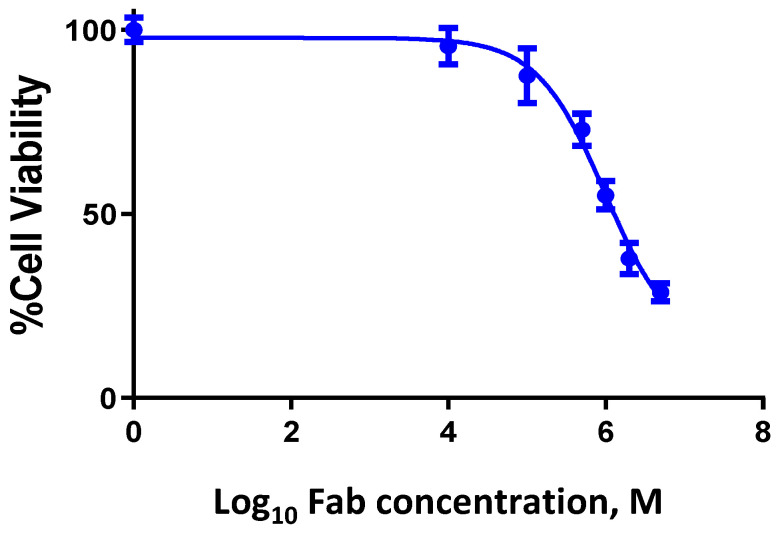
In vitro NT2-D1 cell growth inhibition assays performed with *rh*Fab_3D1 at concentrations ranging between 10 nM and 5 μM. The data account for the average of three independent experiments performed on triplicate wells for each data point.

**Table 1 pharmaceuticals-16-01130-t001:** Theoretical and Experimental Molecular Masses of the Anti-Nodal *rh*Fab_3D1.

rhFab_3D1	Theor. M.W. Da	Exp. M.W. Da
Intact	49,714.54	49,714.63
Light chain	24,279.93	24,280.64
Heavy chain	25,436.60	25,436.83

**Table 2 pharmaceuticals-16-01130-t002:** Kinetic association and dissociation rates and apparent K_D_s derived from the BLI binding experiments of *rh*Fab_3D1 to *rh*Nodal. Only the data obtained at 25 nM, 50 nM, and 100 nM were used to extrapolate the K_D_. The term “Error” given in the table indicates for each individual experiment the point-to-point average deviations calculated between the values of the experimental curves and the values of the curves obtained from fittings. In column 2, the error values refer to the deviations of the K_D_s calculated from the deviations of the corresponding kon and koff.

Conc. (nM)	K_D_ (M)	K_D_ Error	k_a_ (1/Ms)	k_a_ Error	k_dis_ (1/s)	k_dis_ Error	Full R2
6.25	1.00 × 10^−12^	<1.00 × 10^−12^	7.72 × 10^4^	4.07 × 10^3^	1.00 × 10^−7^	<1.00 × 10^−7^	0.9928
12.5	1.54 × 10^−12^	2.15 × 10^−12^	3.14 × 10^4^	8.54 × 10^2^	1.00 × 10^−7^	<1.00 × 10^−7^	0.9986
25	1.12 × 10^−8^	1.21 × 10^−10^	2.15 × 10^4^	2.31 × 10^2^	2.39 × 10^−4^	5.18 × 10^−7^	0.9996
50	6.61 × 10^−9^	4.97 × 10^−11^	6.76 × 10^4^	4.26 × 10^2^	4.47 × 10^−4^	1.84 × 10^−6^	0.9943
100	4.89 × 10^−9^	5.30 × 10^−11^	1.64 × 10^5^	1.37 × 10^3^	8.04 × 10^−4^	5.60 × 10^−6^	0.9240

## Data Availability

Data are available from the authors in their respective laboratories.

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
