# Peer review of "Production in Bacteria and Characterization of Engineered Humanized Fab Fragment against the Nodal Protein"

_pharmaceuticals, 2023, doi:10.3390/ph16081130_

Round 1

Reviewer 1 Report

This paper describes the humanisation and biophysical characterisation of a murine anti-nodal monoclonal antibody. The antibody isolation was described in a previous paper, and this paper describes the characterisation of the humanised version of the antibody. The antibody was humanised using an online tool called Tabhu. This online tool was unfortunately not available at the time of this review – it is unknown whether it is permanently or temporarily unavailable. The website provided in the reference for Tabhu is www.biocomputing.it/tabhu which is no longer available.  If there is an undated website, then please provide a link to the online tool in the methods.

Since the main novelty of the work described in this paper is the production of a humanised version of the antibody, it would be ideal to show some immunogenicity data. There is no evidence that the humanisation has been successful. The authors have shown that the antibody still binds to its target (nodal) and has efficacy in cell-killing assays, but there is no assessment of the immunogenicity to show the effectiveness of the humanisation process. 

The BLI results for the humanised 3D1 Fab fragment are compared with the murine 3D1 whole antibody. It is unknown therefore whether the decrease in affinity is due to the humanisation process or due to the loss of avidity of the monomeric Fab versus dimeric mAb. Similarly for the cytotoxicity assay, the humanised Fab fragment was compared with a different whole antibody, so it is difficult to draw conclusions on whether the humanisation has affected the affinity.

Line 106:  Explain the reasoning for choosing trastuzumab as the template human mab.  Is it the most similar framework sequence to the 3D1 murine antibody? Was this the template suggested by the Tabhu online tool?

Line 128-129 and Figure 2 legend:  Use the term ‘non-reducing’ rather than ‘native’. SDS-PAGE is denaturing due to heat and SDS, therefore it is not native.

Figure 4, Table 2:  Why have the kinetic constants been calculated for each analyte concentration, rather than performing a global fit across all concentrations?  The KD vales vary considerably across the various analyte concentrations, while they should be constant. The sensorgram curves are very linear for the lower analyte concentrations, suggesting that the sensor surface has been immobilised at too high density. If the surface density is too high, true kinetics cannot be measured as there is little dissociation of the analyte (evidenced by the linearity in the association and dissociation phases).

Line 312: Reference 29 cited, but should be Reference 30 – please check all references are corrected cited.

Line 314: The authors have stated ‘Results not shown’ for the output of the humanisation using the software Tabhu. Given that the novelty of this paper is around the humanisation of the anti-nodal antibody, it would be useful to give more details about the humanisation process, including the results of the process (even if in a supplementary section). This would be particularly useful since the Tabhu online server appears to no longer be available (it was not available at the date of this review), so readers are unable to try the tool to see the types of outputs generated.

Moderate revision of grammar is required throughout the manuscript. The abstract requires more revision than the rest of the document.

Reviewer 2 Report

The manuscript submitted by Annamaria Sandomenico and the co-authors is devoted to the analysis of anti-Nodal Fab produced in E. coli. The manuscript has some minor issues, which the reviewer suggests resolving before the manuscript can be recommended for publication.

1) Fab is "Fragment antigen binding", since that "Fab fragment" is not a correct spelling.

2) From the manuscript, it is not obvious how the disulfide bonds were generated in the Fab. Whether they were formed in E. coli during post-translational modification? Anyway, this should be explained in the manuscript 

3) Abbreviations like "CD and SPR" should not be used in the Abstract 

4) Text in lines 42-93 could be divided into several paragraphs, which makes it easier for understanding by readers

5) Data presented in Fig.2a is indistinguishable at resolution 100-200%

6) Line 138: "Bio-Rad", not Biorad

7) Line 179: whether kon and koff should contain subscripted letters? The same is relevant for KD (lines 180-181)

8) What is mean under "Error" in Table 2? This should be described in 2.4 of the paper. 

9) "MTG" in lines 194-209 should be deciphered

10) What does "Log Conc" means (Fig. 6)? 

11) The short protocol and experimental conditions for LC-ESI-TOF-MS should be added to the 4.8 

12) "In vitro" is written in italics and not in different parts of the manuscript. 

13) Caption to Fig SM4 contains squares instead of digits. 

14) What is the physiological concentration of anti-Nodal antibodies which can be achieved in the blood plasma after monoclonal IgG injection? Whether the concentrations studied in 2.6 are equivalent?

15) For the full picture, the risks associated with the use of a recombinant protein expressed in bacteria should be discussed in the text of the paper. I. e. potential for production of antibodies against the Fab, low half-life compared to full-length antibodies, lack of glycosylation and other post-translational modifications, and so on

Sincerely,

Spellchecking by native speaker is recommended 

Author Response

Please see the attchment

Reviewer 3 Report

The content of this paper was to report the production in bacteria and characterization of engineering humanized Fab fragments against the Nodal protein. The experimental design and testing methods in the section of Materials and Methods were detailed and clear. However, the content of the section of Results needs to be enhanced and improved.

1.     Line 99-101, the authors claimed that “ Bioconjugation tests have also been successfully performed to confirm the protein ability to be site-specifically modified through a transglutamination reaction, which is propaedeutic to the anchorage of other suitable drugs.”. However, no sufficient evidence or explanation to support this inference.

2 . In Table 2. Kinetic association and dissociation rates and apparent KDs derived from the BLI binding experiments of rhFab_3D1 to rhNodal. What is the model or equation to explain this data?

3.     In Figure 6, what was the model or theory to explain these data and then support the authors’ conclusion?

4.     The grammar and style of English need to be improved. Many mistakes were found.

Extensive editing of English language required

Round 2

Reviewer 1 Report

I am satisfied with the responses that the authors gave to my suggestions. I understand that due to the cost and complexity of immunogenicity studies, it is not feasible to consider such experiments at this early stage.

It is unfortunate, but outside of the authors' control, that the online tool that was used for humanisation is no longer available. Given this, the paper may be of limited benefit to readers, as they are unable to apply the same procedure; however it may be of interest to see examples of humanisation using freely available online tools (other online tools are available).

All other comments have been addressed in the manuscript.

Improvements have been made to the abstract to improve the grammar.

Reviewer 2 Report

The manuscript has been improved according to the reviewers' comments. The paper can be recommended for publication

Reviewer 3 Report

The content of the revised paper has been improved significantly.

Minor editing of English language required